# DisenBooth: Identity-Preserving Disentangled Tuning for Subject-Driven Text-to-Image Generation

**Hong Chen[1], Yipeng Zhang[1], Simin Wu[3], Xin Wang[1,2]\*,**
**Xuguang Duan[1], Yuwei Zhou[1], Wenwu Zhu[1,2]\***

[1]Department of Computer Science and Technology, Tsinghua University
[2]Beijing National Research Center for Information Science and Technology
[3]Lanzhou University
{h-chen20,zhang-yp22,dxg18,zhou-yw21}@mails.tsinghua.edu.cn
{xin_wang,wwzhu}@tsinghua.edu.cn, wusm21@lzu.edu.cn

## Abstract

Subject-driven text-to-image generation aims to generate customized images of the given subject based on the text descriptions, which has drawn increasing attention. Existing methods mainly resort to finetuning a pretrained generative model, where the identity-relevant information (e.g., the boy) and the identity-irrelevant information (e.g., the background or the pose of the boy) are entangled in the latent embedding space. However, the highly entangled latent embedding may lead to the failure of subject-driven text-to-image generation as follows: (i) the identity-irrelevant information hidden in the entangled embedding may dominate the generation process, resulting in the generated images heavily dependent on the irrelevant information while ignoring the given text descriptions; (ii) the identity-relevant information carried in the entangled embedding can not be appropriately preserved, resulting in identity change of the subject in the generated images. To tackle the problems, we propose DisenBooth, an identity-preserving disentangled tuning framework for subject-driven text-to-image generation. Specifically, DisenBooth finetunes the pretrained diffusion model in the denoising process. Different from previous works that utilize an entangled embedding to denoise each image, DisenBooth instead utilizes disentangled embeddings to respectively preserve the subject identity and capture the identity-irrelevant information. We further design the novel weak denoising and contrastive embedding auxiliary tuning objectives to achieve the disentanglement. Extensive experiments show that our proposed DisenBooth framework outperforms baseline models for subject-driven text-to-image generation with the identity-preserved embedding. Additionally, by combining the identity-preserved embedding and identity-irrelevant embedding, DisenBooth demonstrates more generation flexibility and controllability[1].

## 1 Introduction

Training on billions of text-image pairs, large-scale text-to-image models (Rombach et al., 2022; Saharia et al., 2022; Ramesh et al., 2022) have recently achieved unprecedented success in generating photo-realistic images that conform to the given text descriptions. Thanks to their remarkable generation ability, a more customized generation topic, subject-driven text-to-image generation, has attracted an increasing number of attention in the community (Gal et al., 2022; Ruiz et al., 2022; Wei et al., 2023; Shi et al., 2023; Chen et al., 2023; Kawar et al., 2022; Kumari et al., 2022). Given a small set of images of a subject, e.g., *3 to 5 images of your favorite toy*, subject-driven text-to-image generation aims to generate new images of the same subject according to the text prompts, e.g., *an image of your favorite toy with green hair on the moon*. The challenge of subject-driven text-to-

---

[*]Corresponding Authors.
[1]Our code is available at https://github.com/forchchch/DisenBooth

image generation lies in the requirement that in addition to aligning well with the text prompts, the generated images are expected to preserve the subject identity (Shi et al., 2023) as well.

Existing subject-driven text-to-image generation methods (Gal et al., 2022; Ruiz et al., 2022; Kumari et al., 2022; Dong et al., 2022) mainly rely on finetuning strategy, which finetune the pretrained text-to-image diffusion models (Rombach et al., 2022; Saharia et al., 2022) through mapping the images containing the subject to a special text embedding. However, since the text embedding is designed to align with the given images, information regarding the subject will be inevitably entangled with information irrelevant to the subject identity, such as the background or the pose of the subject. This entanglement tends to impair the image generation in two ways: (i) the identity-irrelevant information hidden in the entangled embedding may dominate the generation process, resulting in the generated images heavily dependent on the irrelevant information while ignoring the given text descriptions, e.g., DreamBooth (Ruiz et al., 2022) ignores the "*in the snow*" text prompts, and overfits to the input image background as shown in row 2 column 4 of Figure 1. (ii) the identity-relevant information carried in the entangled embedding can not be appropriately preserved, resulting in identity change of the subject in the generated images, e.g., DreamBooth generates a backpack with a different color from the input image in row 3 column 4 of Figure 1. Other works (Wei et al., 2023; Shi et al., 2023; Chen et al., 2023; Gal et al., 2023a; Ma et al., 2023) focus on reducing the computational burden, and investigate the problem of subject-driven text-to-image generation without finetuning. They rely on additional datasets that contain many subjects for training additional modules to customize the new subject. Once the additional modules are trained, they can be used for subject-driven text-to-image generation without further finetuning. However, these methods still suffer poor generation ability without considering the disentanglement, e.g., ELITE (Wei et al., 2023) fails to maintain the subject identity in row 2 column 2 and ignores the action "*running*" from the text prompt in row 1 column 2 of Figure 1.

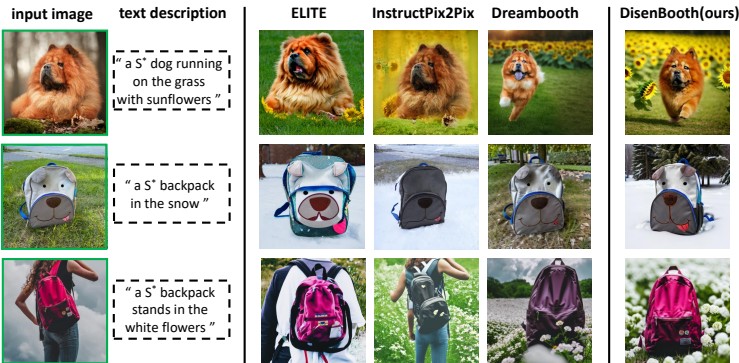

Figure 1: Comparisons between different existing methods and our proposed DisenBooth. *"a S* dog"* is a special token that represents the subject identity. The non-finetuning method ELITE and the image editing method InstructPix2Pix struggle in preserving the subject identity. The existing finetuning method DreamBooth suffers from overfitting on the input image background (row 2 column 4) and subject identity changes (row 3 column 4).

To tackle the entanglement problem in subject-driven text-to-image generation, we propose Disen-Booth, an identity-preserving disentangled tuning framework based on pretrained diffusion models. Specifically, DisenBooth conducts the disentangled tuning during the diffusion denoising process. As shown in Figure 2, different from previous works that only rely on an entangled text embedding as the condition to denoise, DisenBooth simultaneously utilizes a textual identity-preserved embedding and a visual identity-irrelevant embedding as the condition to denoise for each image containing the subject. To guarantee that the textual embedding and the visual embedding can respectively capture the identity-relevant and identity-irrelevant information, we propose two auxiliary disentangled objectives, i.e., the weak denoising objective and the contrastive embedding objective. To further enhance the tuning efficiency, parameter-efficient tuning strategies are adopted. During the inference stage, only the identity-preserved embedding is utilized for subject-driven text-to-image generation. Additionally, through combining the two disentangled embeddings together, we can achieve more flexible and controllable image generation. Extensive experiments show that DisenBooth can si-

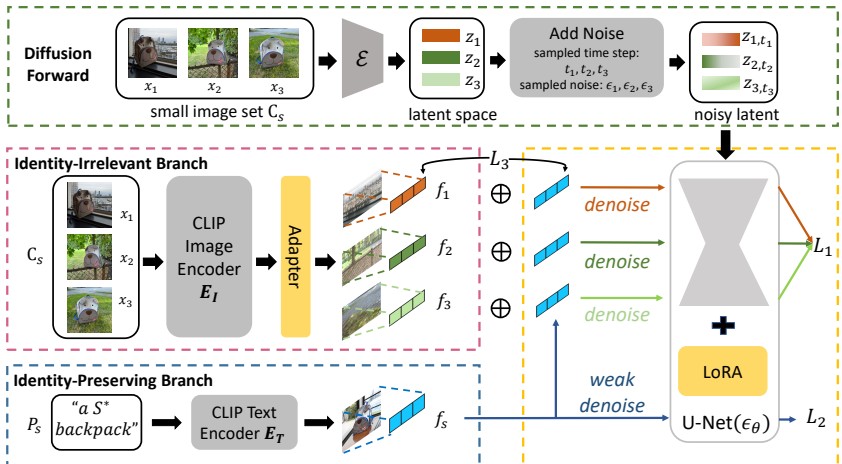

Figure 2: DisenBooth conducts finetuning in the denoising process, where each input image is denoised with the textual embedding $f_s$ shared by all the images to preserve the subject identity, and visual embedding $f_i$ to capture the identity-irrelevant information. To make the two embeddings disentangled, the weak denoising objective $L_2$ and the contrastive embedding objective $L_3$ are proposed. Fine-tuned parameters include the adapter and the LoRA parameters.

multaneously capture the identity-relevant and the identity-irrelevant information, and demonstrates superior generation ability over state-of-the-art methods in subject-driven text-to-image generation.

To summarize, our contributions are listed as follows. (i) To the best of our knowledge, we are the first to investigate the disentangled finetuning for subject-driven text-to-image generation. (ii) We propose DisenBooth, an identity-preserving disentangled tuning framework for subject-driven text-to-image generation, which is able to learn a textual identity-preserved embedding and a visual identity-irrelevant embedding for each image containing the subject, through two novel disentangled auxiliary objectives. (iii) Extensive experiments show that DisenBooth has superior generation ability in subject-driven text-to-image generation over existing baseline models, and brings more flexibility and controllability for image generation.

## 2    RELATED WORK

**Text-to-Image Generation**    Training on large-scale datasets, the text-to-image generation models (Zhang & Agrawala, 2023; Xu et al., 2018; Ramesh et al., 2022; Saharia et al., 2022; Ramesh et al., 2021; Nichol et al., 2022; Chang et al., 2023; Kim et al., 2022) have achieved great success recently. Among these models, diffusion-based models like Stable Diffusion (Rombach et al., 2022), DALLE-2 (Ramesh et al., 2022) and Imagen (Saharia et al., 2022) have attracted a lot of attention due to their outstanding controllability in generating photo-realistic images according to the text descriptions. Despite their superior ability, they still struggle with the more personalized generation, where we want to generate images about some specific or user-defined concepts, whose identities are hard to be precisely described with text descriptions. This leads to the emergence of the recently popular topic, subject-driven text-to-image generation (Ruiz et al., 2022; Gal et al., 2022).

**Text-Guided Image Editing**    Text-guided image editing (Li et al., 2020; Meng et al.; Bar-Tal et al., 2022; Brooks et al., 2022; Kim et al., 2022; Hertz et al., 2022) aims to edit an input image according to the given textual descriptions. SDEdit (Meng et al.)  and Blended-Diffusion (Avrahami et al., 2022) blend the noisy input to the generated image in the diffusion denoising process. Prompt2Prompt (Hertz et al., 2022) combines the attention map of the input image and that of the text prompt to generate the edited image. Imagic (Kawar et al., 2022) utilizes a 3-step optimization-based strategy to achieve more detailed visual edits. A more recent SOTA method InstructPix2Pix (Brooks et al., 2022) utilized GPT-3 (Brown et al., 2020), Stable Diffusion (Rombach et al., 2022) and Prompt2Prompt (Hertz et al., 2022) to generate a dataset with *(original image, text prompt, edited image)* triplets to train a new diffusion model for text-guided image editing. Despite

their effectiveness, they are generally not suitable for subject-driven text-to-image generation (Chen et al., 2023), which needs to perform more complex transformations to the images, e.g., rotating the view, changing the pose, etc. Also, since these methods are not customized for the subject, their ability to preserve the subject identity is not guaranteed. Some examples of InstructPix2Pix for subject-driven text-to-image generation are presented in Figure 1, the pose of the dog is not changed to "*running*" in row 1 column 3, and the subject identity is changed in row 2 and 3.

**Subject-Driven Text-to-Image Generation** Given few images of the subject, subject-driven text-to-image generation (Kumari et al., 2022; Gal et al., 2022; Ruiz et al., 2022; Han et al., 2023) aims to generate new images according to the text descriptions while keeping the subject identity unchanged. DreamBooth (Ruiz et al., 2022) and TI (Gal et al., 2022) are two popular subject-driven text-to-image generation methods based on finetuning. They will both map the images of the subject into a special prompt token $S^*$ during the finetuning process. The difference between them is that TI finetunes the prompt embedding and DreamBooth finetunes the U-Net model. Several concurrent works (Chen et al., 2023; Wei et al., 2023; Shi et al., 2023; Jia et al., 2023; Gal et al., 2023a) propose to conduct subject-driven text-to-image generation without finetuning, which largely reduces the computational cost. They generally rely on additional modules trained on additional new datasets, like the visual encoder in Wei et al. (2023), Shi et al. (2023) to directly map the image of the new subject to the textual space. However, all the existing methods learn the subject embedding in a highly entangled manner, which will easily cause the generated image to have a changed subject identity or to be inconsistent with the text prompt. Although some works (Wei et al., 2023; Dong et al., 2022) use the segmentation mask to exclude the influence of the background, there are some other identity-irrelevant factors that these methods fail to tackle, such as the pose of the subject as shown in row 1 column 2 of Figure 1. Additionally, these methods rely on an additional segmentation model or user-labeled mask, our method is free of additional annotations.

## 3 PRELIMINARIES

In this section, we will introduce the preliminaries about Stable Diffusion and subject-driven text-to-image generation, and also some notations we will use in this paper.

**Stable Diffusion Models.** The Stable Diffusion model is a large text-to-image model pretrained on large-scale text-image pairs $\{(P, x)\}$, where $P$ is the text prompt of the image $x$. Stable Diffusion contains an autoencoder ($\mathcal{E}(\cdot)$, $\mathcal{D}(\cdot)$), a CLIP (Radford et al., 2021) text encoder $E_T(\cdot)$, and a U-Net (Ronneberger et al., 2015) based conditional diffusion model $\epsilon_\theta(\cdot)$. Specifically, the encoder $\mathcal{E}(\cdot)$ is used to transform the input image $x$ into the latent space $z = \mathcal{E}(x)$, and the decoder $\mathcal{D}(\cdot)$ is used to reconstruct the input image from the latent $z$, $x \approx \mathcal{D}(z)$. The diffusion denoising process of Stable Diffusion is conducted in the latent space. With a randomly sampled noise $\epsilon \sim \mathcal{N}(0, I)$ and the time step $t$, we can get a noisy latent code $z_t = \alpha_t z + \sigma_t \epsilon$, where $\alpha_t$ and $\sigma_t$ are the coefficients that control the noise schedule. Then the conditional diffusion model $\epsilon_\theta$ will be trained with the following objective for denoising (Ho et al., 2020; Song et al., 2020):

$$\min \ \mathbb{E}_{P,z,\epsilon,t}[||\epsilon - \epsilon_\theta(z_t, t, E_T(P))||_2^2]. \tag{1}$$

The goal of the conditional model $\epsilon_\theta(\cdot)$ in Eq.(1) is to predict the noise by taking the noisy latent $z_t$, the text conditional embedding obtained by $E_T(P)$, and the time step $t$ as input.

**Finetuning for Subject-Driven Text-to-Image Generation.** Denote the small set of images of the specific subject $s$ as $\mathbb{C}_s = \{x_i\}_{i=1}^K$, where $x_i$ means the $i^{th}$ image and $K$ is the image number, usually 3 to 5. Previous works (Gal et al., 2022; Ruiz et al., 2022; Kumari et al., 2022) will bind a special text token $P_s$, e.g., "*a $S^*$ backpack*" in Figure 1, to the subject $s$, with the following finetuning objective:

$$\min \ \mathbb{E}_{z=\mathcal{E}(x),x \sim \mathbb{C}_s,\epsilon,t}[||\epsilon - \epsilon_\theta(z_t, t, E_T(P_s))||_2^2]. \tag{2}$$

Different methods use the objective in Eq. (2) to finetune different parameters, e.g., Dream-Booth (Ruiz et al., 2022) will finetune the U-Net model $\epsilon_\theta(\cdot)$, while TI (Gal et al., 2022) finetunes the embedding of $P_s$ in the CLIP text encoder $E_T(\cdot)$. DreamBooth finetunes more parameters than TI and achieves better subject-driven text-to-image generation ability. However, these methods bind $P_s$ to several images $\{x_i\}$, making the textual embedding $E_T(P_s)$ inevitably entangled with information irrelevant to the subject identity, which will impair the generation results.

To tackle the problem, in this paper, we propose DisenBooth, an identity-preserving disentangled tuning framework for subject-driven text-to-image generation, whose framework is presented in Figure 2. DisenBooth utilizes the textual embedding $E_T(P_s)$ and a visual embedding to denoise each image. Then, with our proposed two disentangled objectives, the text embedding $E_T(P_s)$ can preserve the identity-relevant information and the visual embedding can capture the identity-irrelevant information. During generation, by combining $P_s$ with other text descriptions, DisenBooth can generate images that conform to the text while preserving the identity. DisenBooth can also generate images that preserve some characteristics of the input images by combining the textual identity-preserved embedding and the visual identity-irrelevant embedding, which provides a more flexible and controllable generation. Next, we will describe how DisenBooth obtains the disentangled embeddings, how DisenBooth finetunes the model with the designed disentangled auxiliary objectives, and then how it conducts subject-driven text-to-image generation with the finetuned model.

## 4 THE PROPOSED METHOD: DISENBOOTH

### 4.1 THE IDENTITY-PRESERVED AND IDENTITY-IRRELEVANT EMBEDDINGS

DisenBooth will use a textual embedding to preserve the identity-relevant information and a visual embedding to capture the identity-irrelevant information. Then, a better subject-driven text-to-image generation can be conducted with the textual identity-preserved embedding.

**The Identity-Preserved Embedding.** We obtain the identity-preserved embedding $f_s$ shared by $\{x_i\}$ through the Identity-Preserving Branch as shown in Figure 2, where we want to map the identity of subject $s$ to a special text token $P_s$. Then the textual identity-preserved embedding can be obtained through the CLIP text encoder,

$$f_s = E_T(P_s). \tag{3}$$

**The Identity-Irrelevant Embedding.** To extract the identity-irrelevant embedding of each image $x_i$, we design an Identity-Irrelevant Branch in Figure 2, where the pretrained CLIP image encoder $E_I$ is adopted, and we can first obtain a feature $f_i^{(p)} = E_I(x_i)$. However, this feature obtained from the pretrained image encoder may contain the identity information, while we only need the identity-irrelevant information in this embedding. To filter out the identity information from $f_i^{(p)}$, we design a learnable mask $M$ with the same dimension as the feature, whose element values belong to (0,1), to filter out the identity-relevant information. Therefore, we obtain a masked feature $M * f_i^{(p)}$, by the element-wise product between the mask and the pretrained feature. Additionally, considering that during the Stable Diffusion pretraining stage, the text encoder is jointly pretrained with the U-Net, while the image encoder is not jointly trained, we use the MLP with skip connection to transform $M * f_i^{(p)}$ into the same space as the text feature $f_s$ as follows,

$$f_i = \text{M} * f_i^{(p)} + \text{MLP}(\text{M} * f_i^{(p)}), i = 1, 2, \cdots, K, \tag{4}$$

and in Figure 2, we denote the skip connection with the mask as the Adapter.

### 4.2 TUNING WITH DISENTANGLED OBJECTIVES

With the above extracted identity-preserved and identity-irrelevant embeddings, we can conduct the finetuning with a similar denoising objective in Eq. (2) on the $K$ images in $\mathbb{C}_s$,

$$\mathcal{L}_1 = \sum_{i=1}^{K} ||\epsilon_i - \epsilon_\theta(z_{i,t_i}, t_i, f_i + f_s)||_2^2. \tag{5}$$

As shown in Figure 2, $\epsilon_i$ is the randomly sampled noise for the $i^{th}$ image, $t_i$ is the randomly sampled time step for the $i^{th}$ image, and $z_{i,t_i}$ is the noisy latent of image $x_i$ obtained by $z_{i,t_i} = \alpha_{t_i}\mathcal{E}(x_i) + \sigma_{t_i}\epsilon_i$ as mentioned in Sec. 3. This objective means that we will use the sum of the identity-preserved embedding and the identity-irrelevant embedding as the condition to denoise each image. Since each image has an image-specific identity-irrelevant embedding, $f_s$ does not have to restore the identity-irrelevant information. Additionally, considering that $f_s$ is shared when denoising all the images, it will tend to capture the common information of the images, i.e., the subject identity. However, only utilizing Eq. (5) to denoise may cause a trivial solution, where the visual embedding $f_i$ captures all

the information of image $x_i$, including the identity-relevant information and the identity-irrelevant information, while the shared embedding $f_s$ becomes a meaningless conditional vector. To avoid this trivial solution, we introduce the following two auxiliary disentangled objectives.

**Weak Denoising Objective.** Since we expect that $f_s$ can capture the identity-relevant information instead of becoming a meaningless vector, $f_s$ should have the ability of denoising the common part of the images. Therefore, we add the following objective:

$$\mathcal{L}_2 = \lambda_2 \sum_{i=1}^{K} ||\epsilon_i - \epsilon_\theta(z_{i,t_i}, t_i, f_s)||_2^2. \tag{6}$$

In this objective, we expect only with the identity-preserved embedding, the model can also denoise each image. Note that we add a hyper-parameter $\lambda_2 < 1$ for this denoising objective, because we do not need $f_s$ to precisely denoise each image, or $f_s$ will again contain the identity-irrelevant information. Combining this objective and the objective in Eq. (5) together, we can regard the process in Eq. (5) as a precise denoising process, and regard the process in Eq. (6) as a weak denoising process. The precise denoising process with $f_s + f_i$ as the condition should denoise both the subject identity and some irrelevant information such as the background, while the weak denoising process with $f_s$ as the condition only needs to denoise the subject identity, so it requires a smaller regularization weight $\lambda_2 < 1$. We use $\lambda_2 = 0.01$ for all our experiments.

**Contrastive Embedding Objective.** Since we expect $f_s$ and $f_i$ to capture disentangled information of the image $x_i$, the embeddings $f_s$ and $f_i$ should be contrastive and their similarities are expected to be low. Therefore, we add the contrastive embedding objective as follows,

$$\mathcal{L}_3 = \lambda_3 \sum_{i=1}^{K} cos(f_s, f_i). \tag{7}$$

Minimizing the cosine similarity between $f_s$ and $f_i$ will make them less similar to each other, thus easier to capture the disentangled identity-relevant and identity-irrelevant information of $x_i$. $\lambda_3$ is a hyper-parameter which is set to 0.001 for all our experiments. Therefore, the disentangled tuning objective of DisenBooth is the sum of the above three parts:

$$\mathcal{L} = \mathcal{L}_1 + \mathcal{L}_2 + \mathcal{L}_3. \tag{8}$$

**Parameters to Finetune.** In previous works, DreamBooth (Ruiz et al., 2022) finetunes the whole U-Net model and achieves better subject-driven text-to-image generation performance. However, DreamBooth requires higher computational and memory cost during finetuning. To reduce the cost while still maintaining the generation ability, we borrow the idea of LoRA (Hu et al., 2021) to conduct parameter-efficient finetuning. Specifically, for each pretrained weight matrix $W_0 \in \mathbb{R}^{d \times k}$ in the U-Net $\epsilon_\theta(\cdot)$, LoRA inserts a low-rank decomposition to it, $W_0 \leftarrow W_0 + BA$, where $B \in \mathbb{R}^{d \times r}, A \in \mathbb{R}^{r \times k}, r \ll min(d, k)$. $A$ is initialized as Gaussian and $B$ is initialized as 0, and during finetuning, only $B$ and $A$ are learnable, while $W_0$ is fixed. Therefore, the parameters to finetune are largely reduced from $d \times k$ to $(d + k) \times r$. Parameters for DisenBooth contain the parameters in the previous adapter and the LoRA parameters in the U-Net, as shown in the yellow block in Figure 2.

## 4.3 MORE FLEXIBLE AND CONTROLLABLE GENERATION

After the above finetuning process, DisenBooth binds the identity of the subject $s$ to the text prompt $P_s$, e.g., "*a S\* backpack*". When generating new images of subject $s$, we can combine other text descriptions with $P_s$ to obtain the new text prompt $P_s'$, e.g., "*a S\* backpack on the beach*". Then, the CLIP text encoder will transform it to its text embedding $f_s' = E_T(P_s')$. With $f_s'$ as the condition, the U-Net model can denoise a randomly sampled Gaussian noise to an image that conforms to $P_s'$ while preserving the identity of $s$. Moreover, if we want the generated image to inherit some characteristics of one of the input images $x_i$, e.g., the pose, we can obtain its visual identity-irrelevant embedding $f_i$ through the image encoder and the finetuned adapter, and then, use $f_s' + \eta f_i$ as the condition of the U-Net model. Finally, the generated image will inherit the characteristic of the reference image $x_i$, and $\eta$ is a hyper-parameter that can be defined by the user to decide how many characteristics can be inherited. DisenBooth not only enables the user to control the generated image by the text, but also by the preferred reference images in the small set, which is more flexible and controllable.

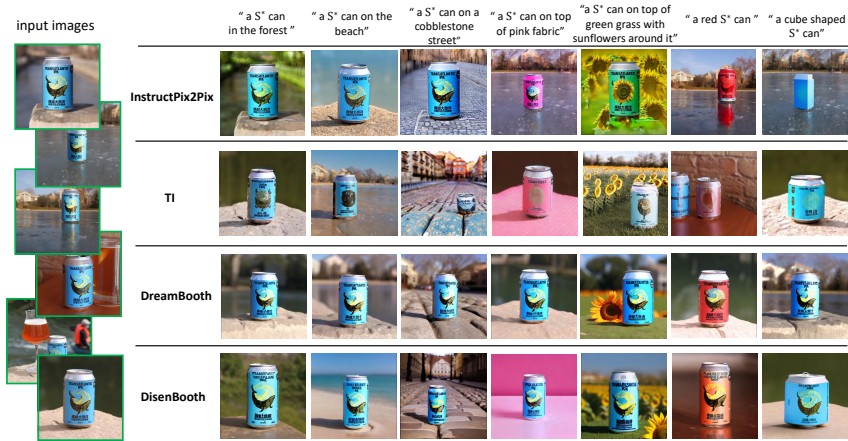

Figure 3: Generated images of the *can* given different text prompts with different methods.

# 5 EXPERIMENTS

## 5.1 EXPERIMENTAL SETTINGS

**Dataset.** We adopt the subject-driven text-to-image generation dataset DreamBench proposed by Ruiz et al. (2022), which are downloaded from Unsplash[2]. This dataset contains 30 subjects, including unique objects like backpacks, stuffed animals, cats, etc. For each subject, there are 25 text prompts, which contain recontextualization, property modification, accessorization, etc. Totally, there are 750 unique prompts, and we follow Ruiz et al. (2022) to generate 4 images for each prompt, and total 3,000 images for robust evaluation.

**Evaluation Metrics.** (i) The first aspect is the subject fidelity, i.e., whether the generated image has the same subject as the input images. To evaluate the subject fidelity, we adopt the DINO score proposed by Ruiz et al. (2022), i.e., the average pairwise cosine similarity between the ViT-S/16 DINO embeddings of the generated images and the input real images. A higher DINO score means that the generated images have higher similarity to the input images, but may risk overfitting the identity-irrelevant information. (ii) The second is the text prompt fidelity, i.e., whether the generated images conform to the text prompts, which is evaluated by the average cosine similarity between the text prompt and image CLIP embeddings. This metric is denoted as CLIP-T (Gal et al., 2022; Ruiz et al., 2022). Besides these two metrics, we also use human evaluation to compare our proposed method and baselines. Specifically, we asked 40 users to rank different methods in their identity-preserving ability, with randomly sampled 15 unique prompts for each user, and we denote the average rank of the 600 ranks as Identity Avg. Rank. Additionally, we asked another 30 users to rank the generated images in the overall performance of the subject-driven text-to-image generation ability, i.e., jointly considering whether the generated images have the same subject as the input images and whether they are consistent with the text prompts. With randomly sampled 30 unique prompts for each user, we finally obtain 900 ranks, and denote this rank as Overall Avg. Rank.

**Baselines.** TI (Gal et al., 2022) and DreamBooth (Ruiz et al., 2022) are finetuning methods for subject-driven text-to-image generation. InstructPix2Pix (Brooks et al., 2022) is a SOTA pretrained method for text-guided text-to-image editing. We also include the pretrained Stable Diffusion (SD) (Rombach et al., 2022) without finetuning as a reference model. We provide the detailed implementation of DisenBooth in A.1.

## 5.2 COMPARISON WITH BASELINES

The scores of different methods are shown in Table 1, and some visualized generated results are shown in Figure 3 and Figure 4. More generated results can be found in the Appendix. From these results, we have the following observations: **(i)** as the reference model, the pretrained SD has the lowest DINO score and the highest CLIP-T score as expected. The pretrained SD is not customized

---

[2]https://unsplash.com/.

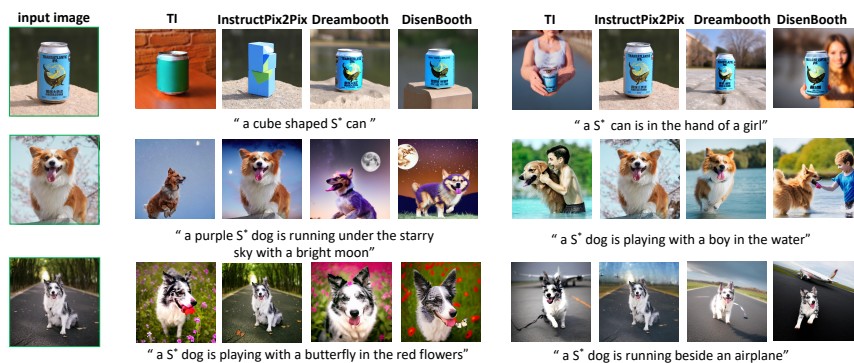

Figure 4: Generated examples of different subjects.

for the subject, thus having the lowest image similarity, but it has the highest CLIP-T score because it does not overfit the input images. **(ii)** The image editing method InstructPix2Pix is not very suitable for the subject-driven text-to-image generation, which has the lowest CLIP-T, i.e., the lowest text prompt fidelity, because it cannot support the complex subject transformations. **(iii)** TI has a weaker ability to maintain the subject identity and a lower prompt fidelity than DreamBooth and DisenBooth. **(iv)** DreamBooth has the highest DINO score, which means that the generated images are very similar to the input images. However, as observed in the generated images, this too high DINO score results from overfitting the identity-irrelevant information, e.g., in Figure 3, the generated images by DreamBooth have a similar background to the last input image, making the prompts like "*on the beach*" and "*on top of pink fabric*" ignored. **(v)** In contrast, our DisenBooth achieves the highest CLIP-T score except the reference model, and generates images that conform to the text prompts while preserving the subject identity. Considering the images generated by our DisenBooth have very different backgrounds from the input images, it has a little lower DINO score than DreamBooth, but it receives a higher Identity Avg. Rank from users, showing its superior ability in preserving the subject identity instead of overfitting the identity-irrelevant information. Additionally, the Overall Avg. Rank collected from the users demonstrates that DisenBooth outperforms existing methods in subject-driven text-to-image generation. We also provide more visualizations in the Appendix, which further shows the superiority of DisenBooth.

Table 1: DINO, CLIP-T, and the user preferences of different methods on DreamBench. Except the referenced model pretrained SD, we bold the method with the best performance w.r.t. each metric.

|  | pretrained SD | InstructPix2Pix | TI | DreamBooth | DisenBooth |
|---|---|---|---|---|---|
| **DINO Score**↑ | 0.362 | 0.605 | 0.546 | **0.685** | 0.675 |
| **CLIP-T Score**↑ | 0.352 | 0.303 | 0.318 | 0.319 | **0.330** |
| **Identity Avg. Rank**↓ | - | 2.69 | 3.73 | 2.20 | **1.37** |
| **Overall Avg. Rank**↓ | - | 2.89 | 3.07 | 2.45 | **1.59** |

### 5.3 ABLATION STUDY

**Disentanglement.** In our design, the textual embedding $f_s$ obtained through the special text prompt $P_s$ aims to preserve the subject identity, and the visual embedding $f_i$ aims to capture the identity-irrelevant information. We verify the relationship in Figure 5. In each row, we generate identity-relevant images only using $f_s$ as the U-Net condition with 4 random seeds. Then, we generate identity-irrelevant images using $f_i$ as the U-Net condition. The results show that DisenBooth can faithfully disentangle the identity-relevant and the identity-irrelevant information. Additionally, in the second row, we can see that the identity-irrelevant information not only contains the background, but also the pose of the dog, which is described with the pose of a human by Stable Diffusion. The disentanglement explains why our DisenBooth outperforms current methods. The shared textual embedding only contains the information about the subject identity, making generating new background, pose or property easier and resulting in better generation results.

**More Flexible and Controllable Generation with $f_i$.** As aforementioned, if we want to inherit some characteristics of the input image $x_i$, we can add the visual embedding $f_i$ to $f'_s$ with a user-defined weight $\eta$, i.e., the condition is $f'_s + \eta f_i$. In Figure 6, with the same text prompt "*a S\* dog on*

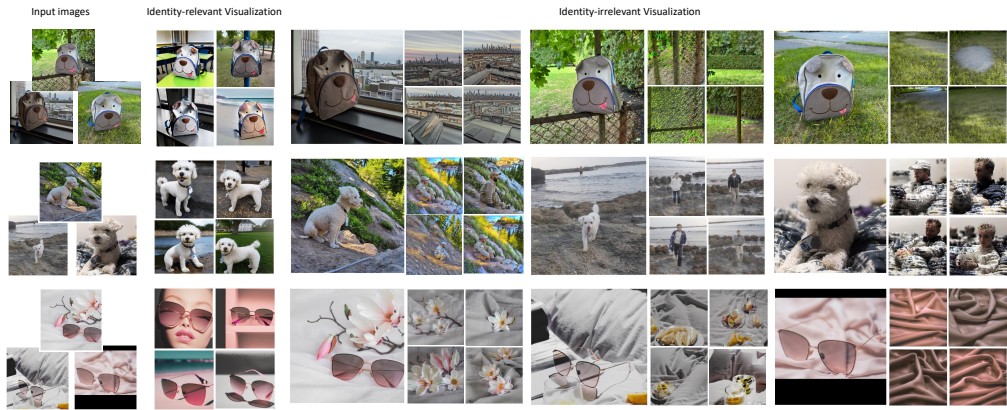

Figure 5: Visualization for disentanglement. The identity-relevant images are generated using the text prompt $P_s$. The identity-irrelevant images are generated with the image-specific identity-irrelevant embedding $f_i$. All generations are with 4 random seeds.

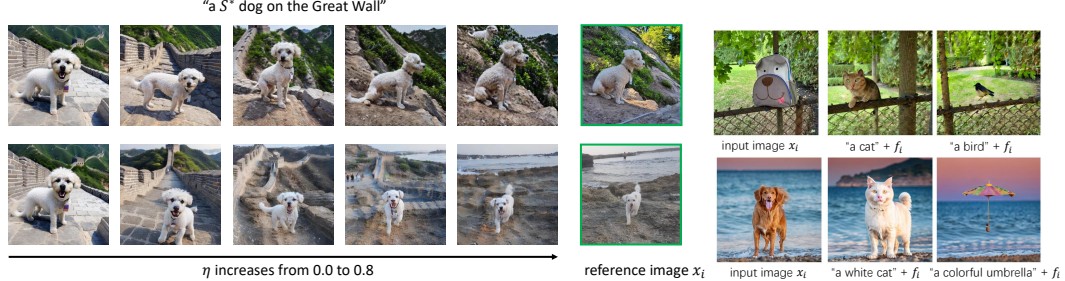

Figure 6: Generating images with different different $\eta$.

Figure 7: Generating other objects with characteristics of $f_i$.

*the Great Wall"* to obtain $f_s'$, we select two different images of the *dog* to obtain $f_i$, and the weight $\eta$ is changed linearly from 0.0 to 0.8. The results show that with larger $\eta$, the generated image will be more similar to the reference image. With a relatively small $\eta$, the generated image can simultaneously conform to the text and inherit some reference image characteristics, which gives a more flexible and controllable generation. However, as $\eta$ becomes large, the text prompt will be ignored and the generated image will be the same as the reference image. This phenomenon means the identity-irrelevant part will impair the subject-driven text-to-image generation, which inspires us to disentangle the tuning process. Additionally, we can also use $f_i$ to generate other images that have the characteristics of the identity-irrelevant information of $x_i$. In Figure 7, we use $f_i$ to generate other objects, such as "a cat". We can see that the generated image inherits the pose and the background of the input image, which further shows the flexibility of our DisenBooth.

We provide more ablation in the Appendix, the effectiveness validation of the disentangled loss in A.2, the mask in A.3, etc., in both qualitative and quantitative ways.

## 6 CONCLUSION

In this paper, we propose DisenBooth for subject-driven text-to-image generation. Different from existing methods which learn an entangled embedding for the subject, DisenBooth will use an identity-preserved embedding and several identity-irrelevant embeddings for all the images in the finetuning process. During generation, with the identity-preserved embedding, DisenBooth can generate images that simultaneously preserve the subject identity and conform to the text descriptions. Additionally, DisenBooth shows superior subject-driven text-to-image generation ability and can serve as a more flexible and controllable framework.

ACKNOWLEDGEMENT

This work was supported by the National Key Research and Development Program of China No. 2023YFF1205001, National Natural Science Foundation of China (No. 62250008, 62222209, 62102222), Beijing National Research Center for Information Science and Technology under Grant No. BNR2023RC01003, BNR2023TD03006, and Beijing Key Lab of Networked Multimedia.

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

## A APPENDIX

### A.1 IMPLEMENTATION DETAIL

We implement DisenBooth based on the Stable Diffusion 2-1 (Rombach et al., 2022). The learning rate is 1e-4 with the AdamW (Loshchilov & Hutter, 2018) optimizer. The finetuning process is conducted on one Tesla V100 with batch size of 1, while the finetuning iterations are ~3,000. As for the LoRA rank, we use $r = 4$ for all the experiments. The MLP used in the adapter is 2-layer with ReLU as the activation function. The LoRA and adapter make the parameters to finetune about $2.9M$, which is small compared to the whole $865.9M$ U-Net parameters. Additionally, the special token $P_s$ we use to obtain the identity-preserved embedding is the same as that of (Ruiz et al., 2022), i.e., "$a + S^* + class$", where $S^*$ is a rare token and $class$ is the class of the subject. When comparing with the baselines, we only use the textual embedding $f'_s$ mentioned in Sec. 4.3 as the condition.

### A.2 THE EFFECTIVENESS OF THE DISENTANGLED OBJECTIVES.

We validate the effectiveness of the proposed two disentangled objectives in Table 2, on half of the DreamBench dataset. We can see that without the weak denoising objective (L2), the DINO score will decrease, which means the subject identity cannot be well preserved. The contrastive embedding objective (L3) can further improve the DINO score and the CLIP-T. We also provide the generated images of the variants in Figure 8, which are consistent with the quantitative results. It can be seen that without the weak denoising objective, the subject identity cannot be well-preserved, e.g., as circled out, the input images have 3 logos but the generated images with this variant have fewer. Without the contrastive embedding objective which makes the textual identity-preserved embedding contain different information from the visual identity-irrelevant embedding, the identity of the subject will be changed when the prompt is "*in the white flowers*" in row 2 column 3. Additionally, in row 2, the generated images seem to have the same angle, which may be entangled in the shared identity-preserved embedding.

|  | w/o weak(L2) | w/o contrast(L3) | w/o both | DisenBooth |
|---|---|---|---|---|
| **DINO↑** | 0.684 | 0.689 | 0.683 | **0.693** |
| **CLIP-T↑** | 0.329 | 0.328 | **0.331** | **0.331** |

Table 2: Ablations about the disentangled objectives.

### A.3 ABLATIONS ABOUT THE MASK

We use a learnable vector $M$ to act as a mask to filter out the identity-relevant information. We compare the performance of our method with and without the mask on half of DreamBench, and the results are shown in Table 3. We can see that without the mask, the DINO score will degrade

|  | w/o mask | DisenBooth |
|---|---|---|
| **DINO↑** | 0.673 | **0.693** |
| **CLIP-T↑** | 0.330 | **0.331** |

Table 3: Ablations about the learnable mask.

significantly. Considering that the mask is used to filter out the identity information, without the mask, the visual branch will also include some identity information, which may cause the text branch to capture less subject identity information, thus resulting in a low DINO score. We also conduct disentanglement visualization about the mask in Figure 9, where we use the learned $f_s$ to generate 4 identity-relevant images about the subject, and use $f_i$ to generate 4 identity-irrelevant images. It can be seen that without the mask, the subject identity cannot be well preserved (e.g., in the identity-relevant images of w/o mask, the color and the logos are different from the input image) and the identity-irrelevant feature will contain identity information (e.g., the identity-irrelevant images of w/o mask will contain the backpack color).

**Comparison between the feature-level mask or pixel-level mask.** In our proposed method, we apply a learnable mask in the adapter to conduct feature selection. A natural question is whether

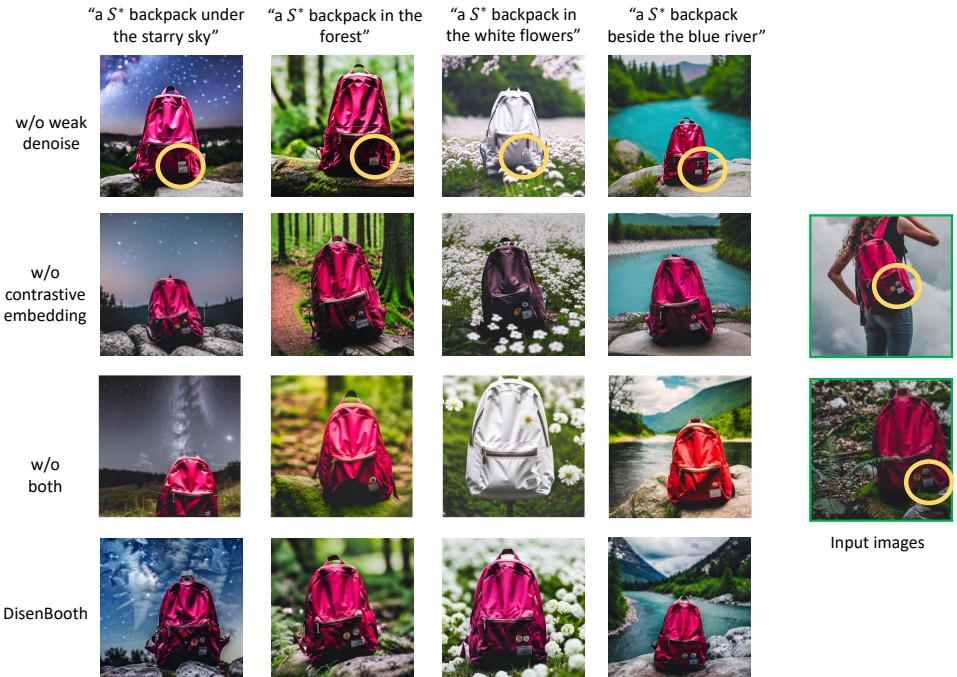

Figure 8: The effectiveness of the disentangled auxiliary objectives.

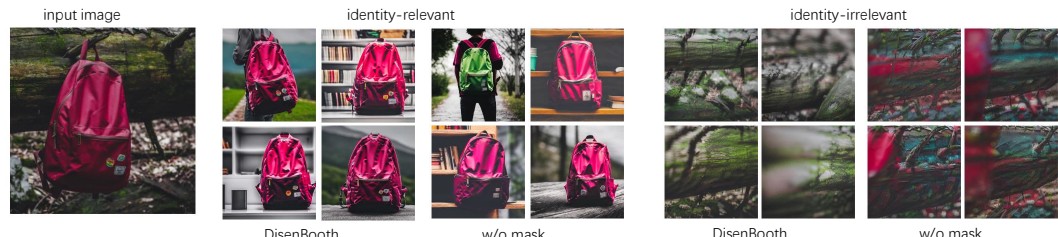

Figure 9: Disentanglement comparison between with and without the mask. Identity-relevant images are generated with the text prompt with 4 random seeds. Identity-irrelevant images are generated with the image-specific identity-irrelevant embedding. The results show that mask M can prevent the identity information into the identity-irrelevant feature, e.g., the red color.

|         | DreamBooth | Pixel Mask | DisenBooth |
|---------|:----------:|:----------:|:----------:|
| **DINO** | **0.671** | 0.666 | 0.666 |
| **CLIP-T** | 0.321 | 0.327 | **0.336** |

Table 4: Comparison between using pixel-level masks or learnable feature-level masks.

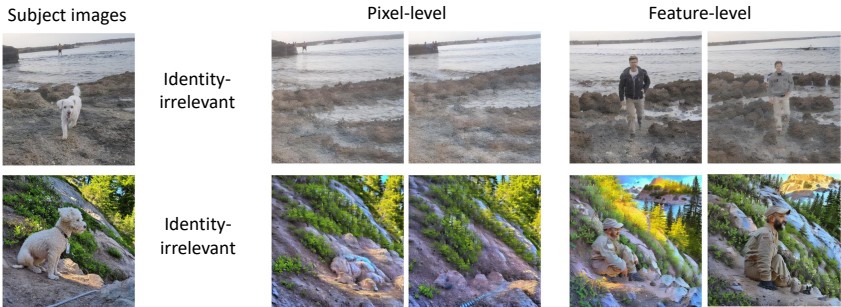

Figure 10: The generated identity-irrelevant image comparison between using the pixel-level mask and feature-level mask.

we can use an explicit pixel-level mask before the image encoding. We compare the results of using pixel-level masks and feature-level masks on 10 subjects in Table 4. We can see that both applying pixel-level masks (Pixel Mask) and feature-level masks (DisenBooth) can improve the CLIP-T score compared to DreamBooth, alleviating overfitting the identity-irrelevant information problem. However, it shows that applying the feature-level mask brings better performance. The reasons are as follows:

- Masks at the feature level filter information at the semantic level instead of the pixel level, enabling it to capture more identity-irrelevant factors such as the pose or position of the subject. Therefore, the identity-preserving branch does not have to overfit these factors, thus achieving a better CLIP-T score and better text fidelity. However, using the masks on the pixels can only disentangle the foreground and the background, ignoring other identity-irrelevant factors such as pose.

- The mask at the feature level is learnable and jointly optimized in the finetuning process, while adding an explicit mask to the image is a two-stage process, whose performance will be a little worse.

In Figure 10, we use the identity-irrelevant features extracted with the feature-level mask and the pixel-level mask to generate images. We can see that when the subject is the dog, the pixel-level mask can only learn the background information of the input subject image. In contrast, our feature-level mask can not only learn the background, but also the pose and the position of the dog, representing it with a human. The qualitative results are consistent with our previous analysis.

A.4 TUNING WITH 1 IMAGE

We compare different methods when there is only 1 image for finetuning for each subject, on 1/3 of the DreamBench dataset. The average quantitative results are presented in Table 5. We can see that our proposed method achieves both the best image and text fidelity, which further shows the superiority of our proposed method in this scenario.

Additionally, we also try to explore whether our method can disentangle the identity-relevant and identity-irrelevant information under this 1-image setting. The corresponding results are presented in Figure 11. We can see that, under the 1-image finetuning setting, the identity-relevant and identity-irrelevant information can still be disentangled by our method. For example, in the first subject customization, we can disentangle the beach and the duck toy. In the fourth example, we can disentangle the cat, and its background and pose.

|          | pretrain_SD | Pix2Pix | TI    | DreamBooth | DisenBooth |
|----------|-------------|---------|-------|------------|------------|
| **DINO** | 0.352       | 0.613   | 0.538 | 0.610      | **0.617**  |
| **CLIP-T** | 0.351     | 0.302   | 0.288 | 0.297      | **0.321**  |

Table 5: Comparison among different methods when tuning with only 1 image for each subject. Except for the reference pretrain_SD baseline, we bold the method with the best performance.

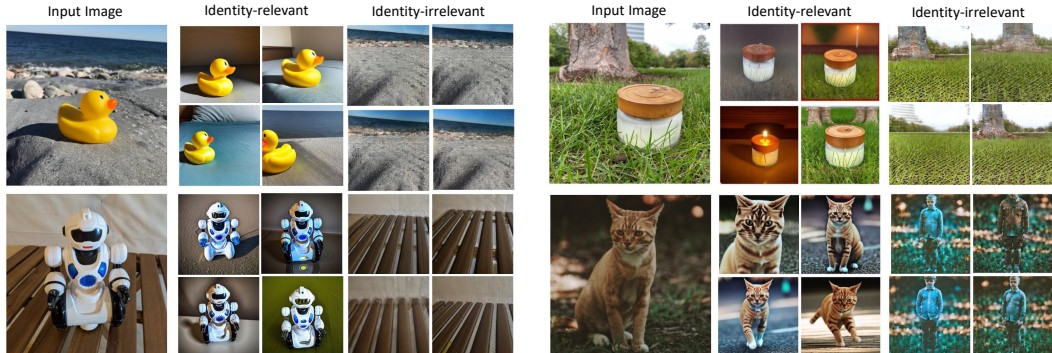

Figure 11: The disentanglement achieved by DisenBooth using only 1 image for finetuning.

### A.5 INFLUENCE OF FINETUNING TEXT ENCODER

We also try to explore the influence of finetuning the CLIP text encoder, where we both finetune the text encoder of DreamBooth and our DisenBooth on 20 subjects in DreamBench. We report the average quantitative results in Table 6. We can see that finetuning the text encoder for both DreamBooth and DisenBooth will increase their DINO score, which means the generated images will be more similar to the input images. However, they suffer a significant CLIP-T drop, which means they overfit the input images while ignoring the given textual prompts. Therefore, finetuning the text encoder will result in more overfitting, but it can be also seen that DisenBooth can alleviate the overfitting problem with a clear margin, whether training the text encoder or not. We also provide the corresponding qualitative comparison in Figure 12, which is consistent with the quantitative results.

### A.6 COMPARE WITH MORE BASELINES

In our main manuscript, we compare DisenBooth mainly with the finetuning method for subject customization. Here, we compare our method with the following more baselines on DreamBench.

CustomDiffusion (short as Custom) Kumari et al. (2022) proposes to only finetune the parameters in the attention layers of the U-Net. SVDiff Han et al. (2023) decomposes the parameters with SVD and only finetunes the corresponding singular values. Break-A-Scene Avrahami et al. (2023) focuses on the scenario where one image may contain several subjects and can customize each subject in the image. ELITE Wei et al. (2023) trains an image-to-text encoder to directly customize each image without further finetuning. E4T Gal et al. (2023b) proposes to train an encoder for each domain and then only very few steps of finetuning are needed for customization.

The corresponding quantitative comparison is presented in Table 7, and the results show that our proposed DisenBooth has a better subject-driven text-to-image generation ability than all the baselines. Note that E4T is a pretrained method and its open-source version is only for human subjects,

|            | DreamBooth | DreamBooth+text | DisenBooth | DisenBooth+text |
|------------|------------|-----------------|------------|-----------------|
| **DINO↑**  | 0.703      | 0.728           | 0.694      | 0.719           |
| **CLIP-T↑**| 0.322      | 0.286           | 0.333      | 0.301           |

Table 6: The influence of finetuning the CLIP text encoder. "+text" means the version of finetuning the text encoder.

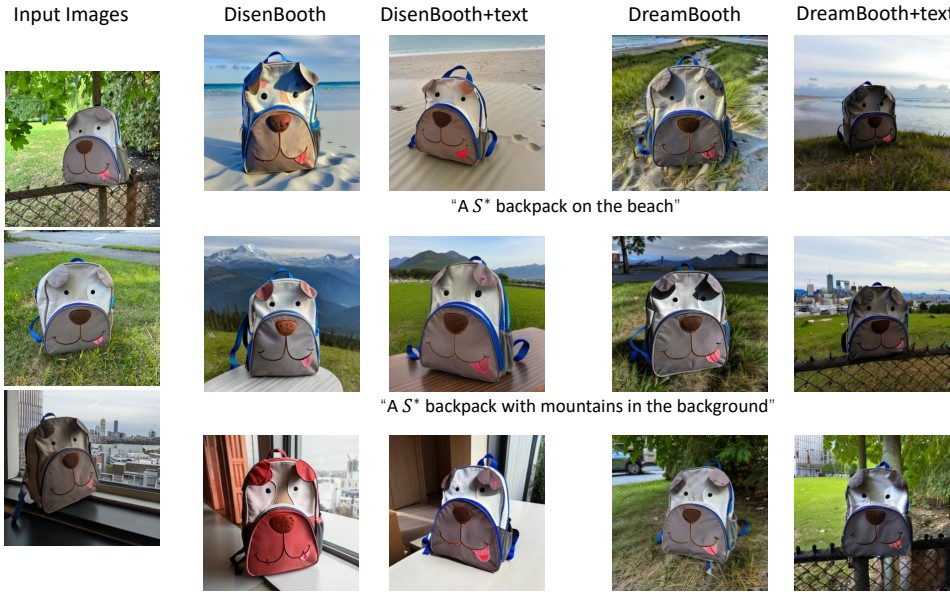

Figure 12: The influence of finetuning the CLIP text encoder.

|  | Custom | SVDiff | Break-A-Scene | E4T | ELITE | DisenBooth |
|---|---|---|---|---|---|---|
| **DINO** | **0.675** | 0.668 | 0.639 | 0.241 | 0.606 | **0.675** |
| **CLIP-T** | 0.314 | 0.318 | 0.313 | 0.262 | 0.291 | **0.330** |

Table 7: Comparison with more baselines on DreamBench.

which faces a severe out-of-domain problem and cannot achieve satisfying performance on Dream-Bench. This phenomenon also indicates that although there are some non-finetuning methods for fast customization, effective finetuning to adapt to new out-of-domain concepts is still important.

We also provide qualitative comparisons with the baselines in Figure 13, which further demonstrates the superiority of our method.

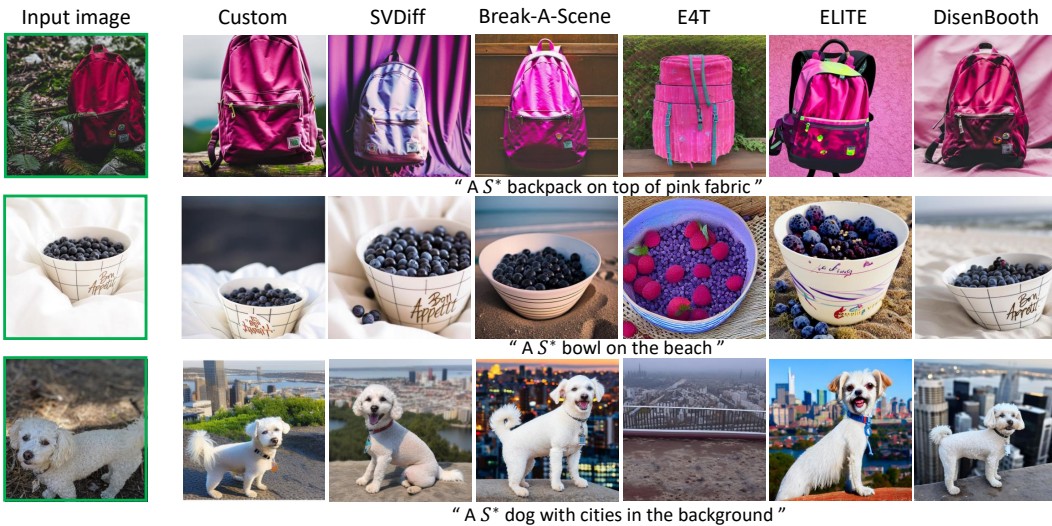

Figure 13: Qualitative comparison with more baselines.

### A.7 Hyper-parameters

There are 2 hyper-parameters in our method, one is the weight of the weak denoising loss, $\lambda_2$, and the other is the weight of the contrastive embedding loss, $\lambda_3$. We use different values of $\lambda_2$ and $\lambda_3$ to tune the model on the "berry_bowl" subject. The results are reported in Table 8 and Table 9. We can see from the results that increasing $\lambda_2$ will make the identity-relevant embedding contain more input image information, which will result in a high DINO score, but faces the problem of overfitting and a low CLIP-T score. Also, increasing $\lambda_3$ will make the embedding disentangled and improve performance, but a too large value like 0.1 will make this objective dominate the optimization process, which will harm the denoising process, and result in a low DINO score and CLIP-T score. Empirically, setting these two parameters to 0.001-0.01 will be OK.

| $\lambda_2$ | 0.0 | 0.001 | 0.01 | 0.1 |
|---|---|---|---|---|
| **DINO** | 0.770 | 0.790 | 0.792 | 0.891 |
| **CLIP-T** | 0.259 | 0.267 | 0.274 | 0.207 |

Table 8: Hyper-parameter experiments on $\lambda_2$

| $\lambda_3$ | 0.0 | 0.001 | 0.01 | 0.1 |
|---|---|---|---|---|
| **DINO** | 0.777 | 0.792 | 0.797 | 0.749 |
| **CLIP-T** | 0.264 | 0.274 | 0.265 | 0.255 |

Table 9: Hyper-parameter experiments on $\lambda_3$

### A.8 Customizing multiple subjects in a scene

We also try to explore whether our DisenBooth can be used when there are multiple subjects in a scene, and the example is shown in Figure 14. As shown on the left of the figure, we first prepare two images for the two subjects, i.e., the duck toy and the cup in the figure, and we want to generate subject-driven images for the two subjects. During disentangled finetuning, most finetuning techniques are the same, except that we will change the input text prompt to "a $S^*$ toy and a $V^*$ cup", which contains the two subjects we are interested in. In the middle and right of the figure, we present the generation results. In the middle, we send the input images to the identity-irrelevant branch to obtain identity-irrelevant features, and generate images using the features with two random seeds. We can see that the generated identity-irrelevant images indeed contain the identity-irrelevant information. On the right, we provide the subject-driven generation results. The first column is the images containing both subjects, where we use "a $S^*$ toy and a $V*$ cup on the snowy mountain/on the cobblestone street" as the prompt. The second column and the third column are about a single subject, where we use "a $S^*$ toy on the snowy mountain/on the cobblestone street" and "a $V^*$ cup on the snowy mountain/on the cobblestone street", respectively.

The preliminary exploration shows that our proposed method can provide overall satisfying generation results on the multi-subject scenario, showing its potential for broader applications. However, looking carefully at the generated images, we can see that although overall the generated cup is similar to the given cup, some visual texture details are a little different. The detail differences may come from that since the identity-relevant branch needs to customize more subjects, and the pixels for each subject become fewer, it is hard for the model to notice every detail of each subject. Beyond the scope of this work, we believe customizing multiple subjects in a scene is an interesting topic and there are more problems worth exploring in future works.

### A.9 Performance on non-center-located images

In the previous examples, we follow previous works Ruiz et al. (2022) to use clear and center-located images of the subject. We also try to see whether our method can still work when the subject only covers fewer pixels and is not center-located. The results are shown in Figure 15. From the results, we can see that the image encoder can still learn the identity-irrelevant information. With the text description and text encoder, the subject can still be generated. Although the generated subject is similar to the given subject, the very few pixels of the subject make it hard to precisely preserve the

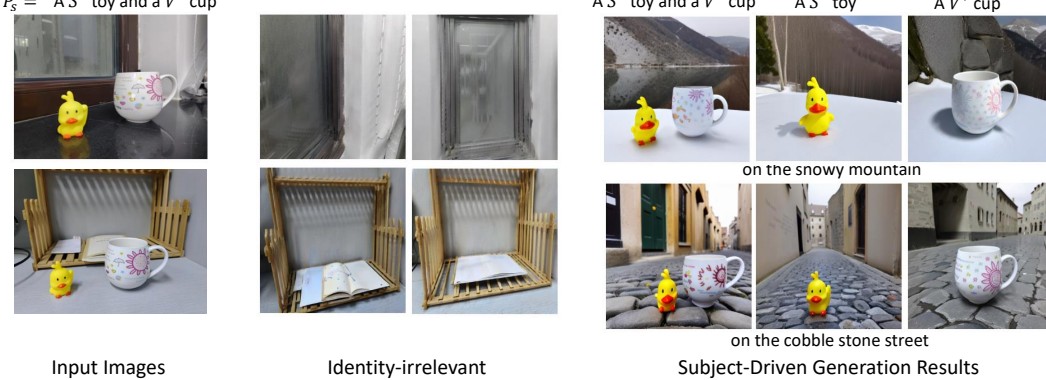

Figure 14: Preliminary exploration on customizing multiple subjects in a scene. The left are the images used for the two subjects, and the text $P_s$ to finetune them. In the middle, we present the generated images using the identity-irrelevant feature as the condition, with two random seeds. On the right, we provide the subject-driven text-to-image generation results, where in the first column we generate images about two subjects with "a $S^*$ toy and a $V^*$ cup", in the second column we generate images about the duck toy with "a $S^*$ toy", and in the third column we generate images about the cup with the prompt "a $V^*$ cup".

details, especially for the second subject. A possible solution to tackle the problem in this scenario is that we can first crop the image and then use a hyper-resolution network to get a clear image for the subject, and then conduct finetuning with DisenBooth.

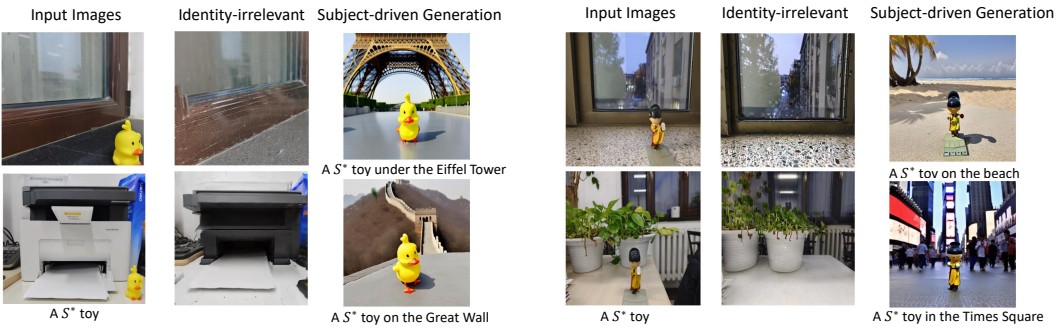

Figure 15: Generation results when the given subject is not located in the center of the image and covers fewer pixels.

### A.10 GENERATION WITH HUMAN SUBJECTS

We also compare our proposed DisenBooth and baselines in generating human subjects. The qualitative results are shown in Figure 16, where we also indicate the used images for each method. For the finetuning baseline TI and DreamBooth, we use 3 images, and E4T only needs a few steps of finetuning with 1 image. ELITE can directly infer without finetuning. For our method, we provide the finetuning results with both 1 image and 3 images. The results show that our method also has clear advantage in both preserving the human identity and conforming to the textual prompt.

### A.11 MORE GENERATION EXAMPLES ON DREAMBENCH

More generation results of different methods on DreamBench are shown in Figure 17, Figure 18, and Figure 19. Denoting the position of the first generated image of InstructPix2Pix as row 1 column 1, from the results, we can observe that:

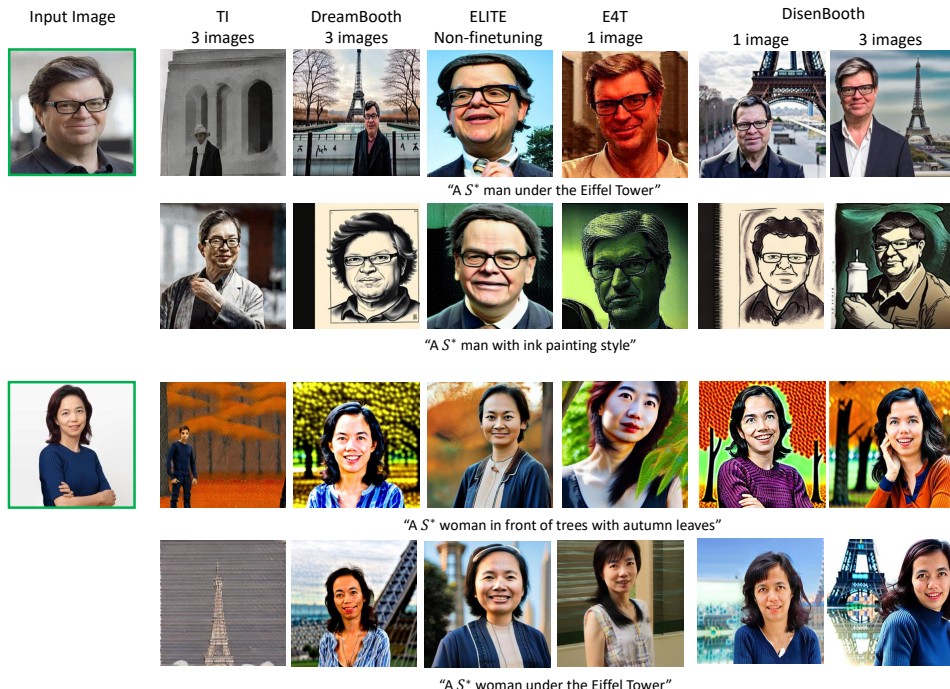

Figure 16: Generation results with human subjects.

- InstructPix2Pix is not suitable for subject-driven text-to-image generation. The generated pictures of InstructPix2Pix always have the same pose as the input images. Additionally, it does not have the idea of the subject, easily making the identity of the subject change. For example, in row 1 column 3 of Figure 17, the color of the backpack is changed to orange. In row 1 column 4 of Figure 18, the color of the can is changed to pink, and in row 1 column 2 of Figure 19, the dog is changed to blue but in fact, we only need a blue house behind the original dog.

- TI usually suffers severe identity change of the subject. For example, in Figure 17, the colors of the backpacks generated by TI are often different from the input images. In Figure 18, in column 3 and column 4 of row 2, the cans also have different identities from the input images.

- DreamBooth suffers from overfitting the identity-irrelevant information in the input images. For example, in Figure 18, almost all the images of the generated cans have the same background as the last input image, making it ignore the text prompts such as *"on the beach"* in row 3 column 2, and *"on top of pink fabric"* in row 3 column 4. Similar phenomenons are also observed in Figure 19, where *"on top of a purple rug"* in row 3 column 3 and *"a purple $S^*$ dog"* in row 3 column 6 are also ignored.

- DisenBooth shows satisfactory subject-driven text-to-image generation results, where the subject identity is preserved and the generated images also conform to the text descriptions.

## A.12 Generation with Anime Subjects

In previous examples, we finetune the subjects on DreamBench. We also use DisenBooth to finetune on some anime characters, and the results are shown in Figure 20. The results show that our DisenBooth works well for these anime subjects.

## A.13 User Study Guidance

We include the guidance for the two user studies. The first one is for the user preference on which method has the best ability to preserve the subject identity. Below is the detailed guidance.

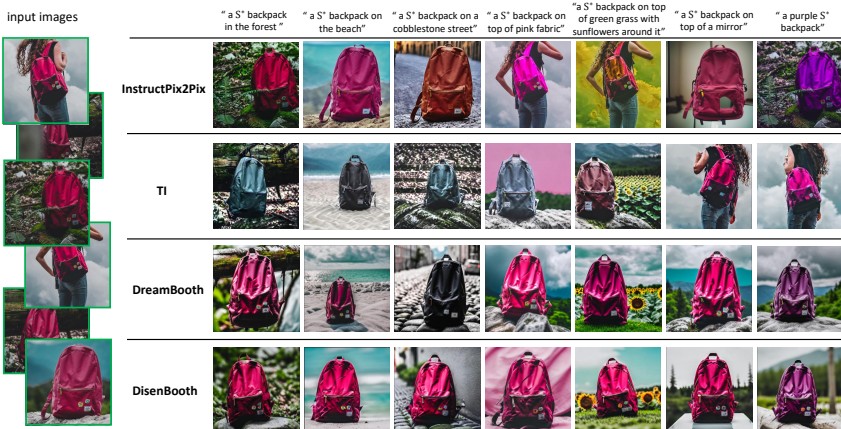

Figure 17: Generated images of the *backpack* given different text prompts with different methods.

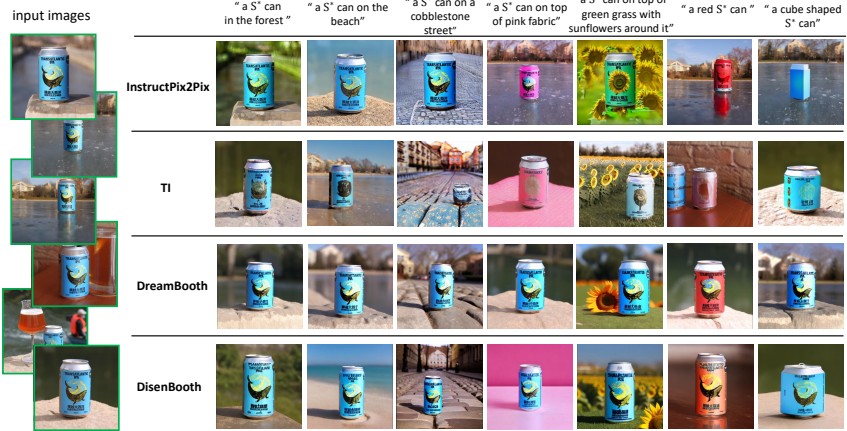

Figure 18: Generated images of the *can* given different text prompts with different methods.

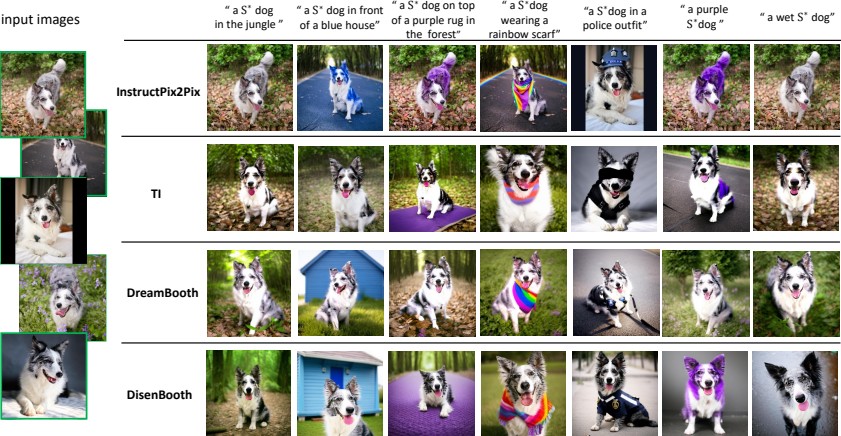

Figure 19: Generated images of the *dog* given different text prompts with different methods.

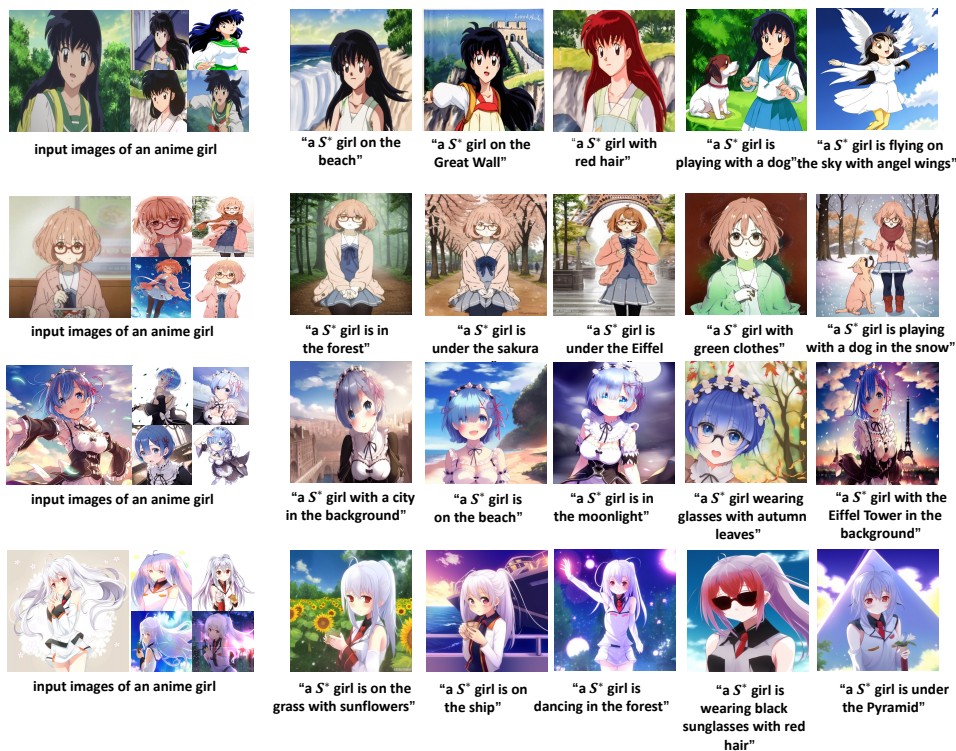

Figure 20: DisenBooth generated examples on some anime subjects.

- Read several reference images of the subject, and keep the subject in mind.

- Rank the given four images. The image that has a more similar subject as the previously given reference images should be ranked top. If two or more images satisfy both requirements, you can give the images with higher quality a top rank.

- During ranking, identity-irrelevant factors like the background and pose should are not considered for similarity.

The second one is for the user preference on which method has the best subject-driven text-to-image generation ability. Below is the detailed guidance.

- Read several reference images of the subject, and keep the subject in mind.

- Read the textual description carefully and then we will show several images generated with the textual description.

- Rank the given four images. The image that has a more similar subject as the previously given reference images and conforms better to the text descriptions should be ranked top. If two or more images satisfy both requirements, you can give the images with higher quality a top rank.

## A.14 LIMITATION DISCUSSION

The limitations of DisenBooth lie in the following two aspects. The first one is that since our DisenBooth is a finetuning method on pretrained Stable Diffusion, it inherits the limitations of the pretrained Stable Diffusion. The second limitation is that since our proposed method does not require any additional supervision, it can only disentangle the subject identity and identity-irrelevant information. How to conduct more fine-grained disentanglement in the identity-irrelevant information, e.g., the pose, the background, and the image style, to achieve a more flexible and controllable generation is worth exploring in the future.

## A.15 OTHER RELATED WORKS

**Disentangled Representation Learning** Disentangled representation learning Wang et al. (2022) aims to discover the explainable latent factors behind data, which has been applied in various fields such as computer vision Lee et al. (2018); Gonzalez-Garcia et al. (2018), recommendation Wang et al. (2021); Chen et al. (2021); Wang et al. (2023), and graph neural networks Zhang et al. (2023; 2022). Learning disentangled representations can not only help to improve the inference explainability, but also makes the model more controllable. In this paper, since we primarily focus on the subject, it is natural for us to disentangle the identity-relevant and identity-irrelevant information for more controllable generation. Particularly, our learned disentangled representations are in a multi-modal fashion, where the identity-relevant information is extracted from the text description and the identity-irrelevant information is extracted from the image.

## A.16 SOCIETAL IMPACTS

Since our work is based on the pretrained text-to-image models, it has a similar societal impact to these base works. On the one hand, this work has great potential to complement and augment human creativity, promoting related fields like painting, virtual fashion try-on, etc. On the other hand, generative methods can be leveraged to generate fake pictures, which may raise some social or cultural concerns. Therefore, we hope that users can judge these factors for using the proposed method in the correct way.

