# OpenReview forum: "DisenBooth: Identity-Preserving Disentangled Tuning for Subject-Driven Text-to-Image Generation"
_ICLR.cc/2024/Conference — ICLR 2024 poster_

### Official Review · Reviewer_QDoi · 2023-10-27

**Soundness:** 4 excellent
**Presentation:** 4 excellent
**Contribution:** 4 excellent
**Rating:** 8
**Confidence:** 4

**Summary:**

The authors propose to disentangle identity-relevant and identity-irrelevant information from the reference set to achieve better identity-preserving performance in personalized text-to-image generation. By guiding the finetuning of the diffusion model denoising process with disentangled textual identity-preserved embedding and visual identity-irrelevant embedding, the personalization results can achieve high fidelity to the target object and flexibility to adapt to the background of any reference image. The disentanglement is constrained by proposed weak denoising and contrastive embedding objectives to ensure a non-trivial solution. Numerical and qualitative performances of the work demonstrate the superiority of the work.

**Strengths:**

- Valid motivation and objective
- Reasonable thoughts of ideas to address the identity-preserving problem in personalization with disentangling feature embeddings
- Thorough experiments

**Weaknesses:**

- Missing reference and comparison [1][2][3]
- Lack of experimental results on real human figures. Since the focus of this work is identity-preserving, it is of great interest to me to see how it works on human objects. Ideally, the authors should provide a comparison figure like Figure 3 in [1].

[1] Rinon Gal, Moab Arar, Yuval Atzmon, Amit H. Bermano, Gal Chechik, Daniel Cohen-Or. Encoder-based Domain Tuning for Fast Personalization of Text-to-Image Models. arXiv preprint arXiv:2302.12228 (2023).
[2] Chen, Wenhu and Hu, Hexiang and Li, Yandong and Ruiz, Nataniel and Jia, Xuhui and Chang, Ming-Wei and Cohen, William W. Subject-driven Text-to-Image Generation via Apprenticeship Learning. NeurIPS 2023.
[3] Xuhui Jia, Yang Zhao, Kelvin C.K. Chan, Yandong Li, Han Zhang, Boqing Gong, Tingbo Hou, Huisheng Wang, Yu-Chuan Su. Taming Encoder for Zero Fine-tuning Image Customization with Text-to-Image Diffusion Models. arXiv preprint arXiv:2304.02642, 2023.

**Questions:**

- Why ELITE is not included in quantitative comparison?
- InstructPix2Pix uses instructions to guide image editing. I wonder how it is used for comparison in this paper.
- A small problem with the structure. Since the finetuning objective consists of several separate losses and two of them are novel ones proposed for disentanglement, the ablation of the disentangled objectives is what readers are most curious about. I think the authors should consider moving Appendix A.1 to the paper section 5.3 if possible.
- Typo: Figure 5 caption line 1. "The identity-irrelevant images are generated using the text prompt Ps." should be "The identity-relevant images"

---

> ### Author Response · Authors · 2023-11-19
> **Response to Reviewer QDoi**
>
> Thank the reviewer for the constructive comments and suggestions. We have revised our paper according to all the reviewers' suggestions and uploaded the revised version, where the revised contents are colored in blue for easy reading. The reviewer can download the revised version for more qualitative results and analysis. We address your concerns point by point as follows.
>
> 1. **Reference and comparisons(W1/Q1)**:
>
>     + We have added all the mentioned references in the related work in our revised version. Thanks for making our related work more complete.
>     + We have also compared the mentioned baselines together with the baselines suggested by Reviewer UqfN in Appendix A.5 in our revised version. Both the qualitative(Table 7) and quantitative results(Figure 13) show the superiority compared to the baselines. For easier reading, we also present the quantitative results in the following table, which shows that our proposed DisenBooth has the best image fidelity and text fidelity.
>     + Note that E4T[1] is a pretrained method and its open-source version is only for human subjects, which faces severe out-of-domain problem and cannot achieve satisfying performance on DreamBench. This phenomenon also indicates that although there are some non-finetuning methods for fast customization, effective finetuning to adapt to new out-of-domain concepts is still important. Although we tried to compare with [2] [3] in the suggested references, unfortunately we found these two works are two pretrained non-finetuning methods by Google, and there are no open-source codes and checkpoints for the pretrained models. We cannot compare with them because we have no access to the pretrained models.
>
>         |        | Custom | SVDiff | Break-A-Scene |   E4T  |  ELITE | DisenBooth(ours) |
>         |--------|:------:|:------:|:-------------:|:------:|:------:|:----------:|
>         |  DINO  | **0.675**  | 0.668  |        0.639  | 0.241  | 0.606  |   **0.675**    |
>         | CLIP-T | 0.314  | 0.318  |        0.313  | 0.262  | 0.291  |   **0.330**    |
>     + Following the reviewer's advice, we have compared with ELITE quantitatively in our revised version and also in the table above. In our first version, we just want to compare our method with the finetuning-based method for fair comparison consideration, while ELITE is a pretrained and non-finetuning one.
>
> 2. **Qualitative results on real human figures(W2)**:
>
>     Following the reviewer's advice, we provided the visual comparison with the baselines on the human subjects in Appendix A.9 and Figure 16 in our revised version. The results further show that our proposed method has a clear advantage over the baselines in both preserving the human identity and conforming to the textual prompt.
>
> 3. **How InstructPix2Pix is used for comparison(Q2)**:
>
>     We use an example to show how we use InstructPix2Pix for subject-driven text-to-image generation as follows. Assuming that we want to customize the backpack and we have an input image of it, we want to generate an image of the backpack under the Eiffel Tower. What we do in the experiment is send the input image to InstructPix2Pix, and then use the instruction ``put the backpack under the Eiffel Tower'' to generate an image. In the DreamBench setting, the backpack has 6 input images, we will repeat the generation process for each input image and calculate the average DINO and CLIP-T score of all the generated images.
>
> 4. **Paper Structure(Q3)**:
>
>     We have tried to move Appendix A.1 to section 5.3, but we find it will exceed the page limit of ICLR. To emphasize this important part, in our revised version, we give the detailed cross-reference at the last of section 5.3 to make the readers easily find this ablation. Thanks for your suggestions.
>
> 5. **Typo(Q4)**:
>
>     We have fixed this typo. Thanks for your careful reading and pointing it out.
>
> Thanks for the valuable suggestions from the reviewer. We think the revised content makes our experiments more comprehensive and our paper more clear.

---

> ### Author Response · Authors · 2023-11-22
> **Gentle reminder**
>
> Dear reviewer, thank you again for the review and we hope that our response and the uploaded revised paper have addressed your concerns. As the author-reviewer is about to end, we would greatly appreciate your feedback and please feel free to let us know if you have any other questions.

---

> > ### Comment · Reviewer_QDoi · 2023-11-22
> >
> > Thank the authors for their carefully prepared rebuttal. The experimental results are comprehensive and the clarifications are nice and clear. I am happy to raise my score.

---

> ### Author Response · Authors · 2023-11-22
> **Thanks to the response**
>
> Thank the reviewer for the response. This is a really fruitful rebuttal process, where the paper is further polished with more clear writing and the results are further strengthened with the suggested experiments. Thanks for your suggestions and response again.

---

### Official Review · Reviewer_UqfN · 2023-10-31

**Soundness:** 3 good
**Presentation:** 3 good
**Contribution:** 3 good
**Rating:** 8
**Confidence:** 4

**Summary:**

This paper proposes the DisenBooth for disentangled and identity-preserving subject-driven text-to-image generation. It learns two distinct embeddings to capture the identity-relevant and identity-irrelevant information, respectively, with a weak denoising loss and contrastive embedding loss to facilitate disentanglement. The experimental results show good qualitative performance.

**Strengths:**

1. The proposed method is simple but effective for disentangled finetuning.
2. The results are good, and extensive ablation experiments are conducted.
3. The paper is well-written and easy to follow.

**Weaknesses:**

1. To exclude the identity-irrelevant information such as background, one straightforward way is to filter them using a subject mask, which can be easily obtained by a segmentation model (e.g., SAM). Recent studies like Break-a-scene[1] have also explored masked diffusion loss for disentangling objects and backgrounds. This paper should also compare with these methods.
2. The proposed method might rely on multiple images to learn the identity-relevant and irrelevant embeddings. When there is only one single image for training, it may be challenging to disentangle them. Although the paper includes experiments on a single image in Section A.3, there is a lack of visual results to verify the effectiveness (like Figure 5), and the number of testing samples is also limited (only 3).
3. More recent methods should be compared, including Break-a-scene[1], Custom Diffusion[2], and SVDiff[3].


[1] Avrahami, Omri, et al. "Break-A-Scene: Extracting Multiple Concepts from a Single Image." In SIGGRAPH Asia 2023.
[2] Kumari, Nupur, et al. "Multi-concept customization of text-to-image diffusion." In CVPR 2023.
[3] Han, Ligong, et al. "Svdiff: Compact parameter space for diffusion fine-tuning." In ICCV 2023.

**Questions:**

See above weaknesses.

---

> ### Author Response · Authors · 2023-11-19
> **Response to Reviewer UqfN**
>
> Thank the reviewer for the constructive comments and suggestions. We have revised our paper according to all the reviewers' suggestions and uploaded the revised version, where the revised contents are colored in blue for easy reading. The reviewer can download the revised version for more qualitative results and analysis. We address your concerns point by point as follows.
>
> 1. **Comparison with more methods(W1/W3)**:
>
>     Following the reviewer's advice, we compare DisenBooth to the mentioned baselines together with the baselines suggested by Reviewer QDoi in Appendix A.5 in our revised version, where the reviewer can see both quantitative (Table 7) and qualitative comparisons (Figure 13) on the DreamBench. Both the qualitative and quantitative results show the superiority compared to the baselines.
>
>     For easier reading, we also present the quantitative results in the following table, which shows that our proposed DisenBooth has the best image fidelity and text fidelity.
>     Note that E4T is a pretrained method and its open-source version is only for human subjects, which faces severe out-of-domain problem and cannot achieve satisfying performance on DreamBench. This phenomenon also indicates that although there are some non-finetuning methods for fast customization, effective finetuning to adapt to new out-of-domain concepts is still important.
>
>     |        | Custom | SVDiff | Break-A-Scene |   E4T  |  ELITE | DisenBooth(ours) |
>     |--------|:------:|:------:|:-------------:|:------:|:------:|:----------:|
>     |  DINO  | **0.675**  | 0.668  |        0.639  | 0.241  | 0.606  |   **0.675**    |
>     | CLIP-T | 0.314  | 0.318  |        0.313  | 0.262  | 0.291  |   **0.330**    |
>
> 2. **More experiments for one single image finetuning(W2)**:
>
>     Following the reviewer's advice, we have conducted experiments of tuning with only 1 image on more subjects (1/3 of the DreamBench dataset), and we give both quantitative results(Table 5) and qualitative visual results(Figure 11) in Appendix A.3 in our revised version. The results show that our proposed method can still disentangle under this 1-shot challenging scenario. For easier reading, we give the average quantitative results in the following table, which shows the superiority over existing methods.
>
>     |        | Pix2Pix |   TI   | DreamBooth | DisenBooth(ours) |
>     |--------|:-------:|:------:|:----------:|:----------:|
>     |  DINO  |  0.613  | 0.538  |   0.610    |   **0.617**    |
>     | CLIP-T |  0.302  | 0.288  |   0.297    |   **0.321**    |
>
> Thanks for your suggestions, and we believe the revised contents further support our work and improve the quality of our paper.

---

> ### Author Response · Authors · 2023-11-22
> **Gentle reminder**
>
> Dear reviewer, thank you again for the review and we hope that our response and the uploaded revised paper have addressed your concerns. As the author-reviewer is about to end, we would greatly appreciate your feedback and please feel free to let us know if you have any other questions.

---

> > ### Comment · Reviewer_UqfN · 2023-11-22
> > **Official Comment by Reviewer UqfN**
> >
> > Thanks the authors for your efforts. After reading the rebuttal and comments from the other reviewers, most of my concerns have been addressed. Therefore, I would like to raise my score.

---

> > > ### Author Response · Authors · 2023-11-22
> > > **Thanks for the response**
> > >
> > > We sincerely thank the reviewer for the response, and we believe the recommended ablation about 1 image and more baselines make our paper more comprehensive. Thanks for your constructive suggestions and response again.

---

### Official Review · Reviewer_9xGd · 2023-10-31

**Soundness:** 3 good
**Presentation:** 3 good
**Contribution:** 3 good
**Rating:** 8
**Confidence:** 2

**Summary:**

The paper focuses on the problem of entanglement of the global and the local (object-specific) features while editing or personalizing a Diffusion model. The argument put forward is that the current state-of-the-art methods used for text-to-image editing tasks in the diffusion domain are not aware of the identity of the object. This means that while editing the images, the background information might dominate the editing process, ignoring the foreground object, or the identity of the object may be compromised. The paper argues that this is due to the entangled nature of the information used in the denoising process of these diffusion models. To address this, the paper introduces disentangled embedding derived from CLIP-based image and text encoders to make sure the global and object-centric features are extracted and used in a disentangled manner with the diffusion model. The proposed method, DisenBooth,  further designs the novel weak denoising and contrastive embedding auxiliary tuning objectives to achieve the disentanglement. Throughout the paper, identity-preserving edits are shown and compared quantitatively and qualitatively with the competing methods.

**Strengths:**

1) The paper proposes a way to disentangle the identity of the object used for editing or personalization in diffusion models. This has an impact on personalized editing, where the user does not want the identity of the object to be compromised while the editing is performed. To achieve this, the paper employs identity-preserving embeddings and two losses to ensure that the foreground information is disentangled from the background information while denoising the images.

2) The paper shows the results of their method in different scenarios. The supplementary shows results on the anime dataset. The quality of the results is impressive compared to the competing methods.

3) The paper also conducts a thorough ablation study of the components used in the method. The study shows the contribution of loss terms and the mask affect the efficiency of the said method. The paper also shows editing scenarios where only one image is used for the personalization task. The results show that the proposed method is better than the competing method in this scenario as well.

**Weaknesses:**

1) The paper employs a LoRA optimization strategy for fine-tuning the diffusion model, but it doesn't address how this choice impacts the quality of their approach compared to fine-tuning the entire UNet. It raises questions about whether the proposed loss functions and encoding strategies would still be effective in such a scenario and what impact this might have on result fidelity.

2) The paper primarily concentrates on editing and personalizing single, often centrally located objects. While this is a challenging task, it would be intriguing to see how the method performs when dealing with scenes containing multiple objects. It is particularly important to understand how the encoding and disentanglement processes behave in such complex scenarios.

3) It remains uncertain whether the mask could be explicitly integrated into the images before the encoding steps. How this approach might affect performance and how does it compare to the current masking strategy employed in the adaptor?

4) The method appears to primarily focus on objects placed at the center of an image. What remains unclear is how it performs when objects are located in the corners of the image or occupy fewer pixels. It's essential to investigate potential failure cases in this context and assess whether the image and text encoders can accurately capture identity-preserving information in such scenarios.

**Questions:**

1) Regarding the choice of LoRA optimization over fine-tuning the entire UNet, how does this affect the quality of the proposed method compared to alternative fine-tuning strategies?

2) While the paper focuses on editing single objects, have the authors explored how the method performs with multiple objects in a scene? What challenges and insights arise when dealing with such complex scenarios, especially in terms of encoding and disentanglement?

3) Concerning the inclusion of masks, could you discuss the potential benefits and drawbacks of explicitly integrating masks into images before encoding? How does this compare to the current masking strategy in the adaptor, and under what circumstances might one approach be more advantageous?

4) The method appears to emphasize objects at the center of an image. Have you tested its performance with objects located in the corners or covering fewer pixels? What are some challenges or failure cases in such scenarios, and how do the image and text encoders adapt to encode identity-preserving information effectively?

---

> ### Author Response · Authors · 2023-11-19
> **Response 1 to Reviewer 9xGd**
>
> Thank the reviewer for the constructive comments and suggestions. We have revised our paper according to all the reviewers' suggestions and uploaded the revised version, where the revised contents are colored in blue for easy reading. The reviewer can download the revised version for more qualitative results and analysis. We address your concerns point by point as follows.
>
> 1. **Lora Finetuning v.s. Full Finetuning(W1/Q1)**:
>
>     Following the reviewer's suggestions, we have conducted experiments on finetuning the U-Net on 10 subjects and compared it with the results of Lora finetuning. The experimental results are provided in the following Table. DisenBooth-full in the Table means we finetune the whole U-Net in DisenBooth.
>
>     |        | DreamBooth | DisenBooth-full | DisenBooth(ours) |
>     |--------|:----------:|:---------------:|:----------:|
>     |  DINO  |   **0.671**    |      0.664      |   0.666    |
>     | CLIP-T |   0.321    |      0.333      |   **0.336**    |
>
>     From the results, we can see that the performance of finetuning the whole U-Net is comparable to Lora finetuning, and additionally, both DisenBooth-full and DisenBooth can alleviate the impact of overfitting and achieve higher CLIP-T(textual fidelity). However, considering that finetuning the whole U-Net for each subject will averagely consume 122 min while finetuning lora only requires 38 min on our V100 32G GPU. Therefore, we recommend finetuning with lora for efficiency.
>
> 2. **Customization for multiple subjects in a scene(W2/Q2)**:
>
>     The reviewer proposes a very interesting scenario to customize multiple subjects in a scene, and we conduct a preliminary experiment under this scenario in Appendix A.7 in our revised version, which the reviewer can refer to for qualitative generation results(Figure 14) and analysis.
>
>     During customizing multiple subjects in a scene, we will use a prompt to describe the multiple subjects for the identity-preserving branch, and leave the identity-irrelevant branch to learn other features. The preliminary exploration results show that our proposed method can also be applied to this scenario and generate overall satisfying images, demonstrating the potential of DisenBooth for broader applications.
>
>     Also, there are some challenges. Although the generated subjects are overall quite similar to the given subjects, some visual texture details of the generated subjects are a little different from the input subjects. The detail differences may result from the following reason: since the identity-preserving branch needs to customize more subjects, and the pixels for each subject become fewer, it will be hard to notice every detail of each subject. Although beyond the scope of this work, we believe customizing multiple subjects in a scene is an interesting topic and there are more problems worth exploring in future works.
>
> 3. **Using masks on pixels or on features(W3/Q3)**:
>
>     Following the reviewer's advice, we compare the results of using pixel-level masks (explicitly integrating masks into images before encoding) and feature-level masks (masking in the adaptor) on 10 subjects in the following Table.
>
>     |        | DreamBooth | Pixel Mask | DisenBooth(ours) |
>     |--------|:----------:|:-------------:|:----------:|
>     |  DINO  |   **0.671**    |     0.666     |   0.666    |
>     | CLIP-T |   0.321    |     0.327     |   **0.336**    |
>
>     We can see that both applying pixel-level masks (Pixel Mask) and feature-level masks (DisenBooth) can improve the CLIP-T score compared to DreamBooth, alleviating overfitting the identity-irrelevant information problem. However, it shows that applying the feature-level mask brings better performance. The reasons are as follows:
>
>     + Masks at the feature level are filtering information at the semantic level instead of pixel level, enabling it to capture more identity-irrelevant factors such as the pose or position of the subject. Therefore, the identity-preserving branch does not have to overfit these factors, thus achieving better CLIP-T score and better text fidelity. However, using the masks on the pixels can only disentangle the foreground and the background, ignoring other identity-irrelevant factors such as pose.
>
>     + The mask at the feature level is learnable and jointly optimized in the finetuning process, while adding an explicit mask to the image is a two-stage process, whose performance will be a little worse.
>
>     We add the quantitative results (Table 4) and also qualitative analysis (Figure 10) in Appendix A.2 in our revised version to better explain the reasons (colored in blue). Generally speaking, if the users do not want to change other identity-irrelevant factors except for the background, such as the pose of the subject during generation, both ways can be applied. Otherwise, applying the mask at the feature level as our method may bring better performance.

---

> ### Author Response · Authors · 2023-11-19
> **Response 2 to Reviewer 9xGd**
>
> 4. **Performance with objects in corners and covering fewer pixels(W4/Q4)**:
>
>     Following the reviewer's suggestions, we conduct some experiments when the objects only cover fewer pixels and are not located in the center. The experimental results are presented in our revised paper in Appendix A.8 and Figure 15. The results show that:
>     + Our method can still effectively learn the identity-preserving and identity-irrelevant information.
>     + However, similar to the multi-subject scenario, when the subject covers very few pixels, although the generated subject is overall similar to the given subject, some visual details are different. Therefore, in this scenario, a possible solution to preserve the details is to first crop the original image and then use a hyper-resolution network to obtain a clear image of the subject. After that, we can conduct finetuning with DisenBooth.
>
> Thanks for your suggestions, we believe the review and the revised content can show more potential of DisenBooth and further improve the quality of the paper.

---

> ### Author Response · Authors · 2023-11-22
> **Gentle reminder**
>
> Dear reviewer, thank you again for the review and we hope that our response and the uploaded revised paper have addressed your concerns. As the author-reviewer is about to end, we would greatly appreciate your feedback and please feel free to let us know if you have any other questions.

---

> > ### Comment · Reviewer_9xGd · 2023-11-22
> >
> > Thanks for the clarifications and experiments. I maintain my score.

---

> > > ### Author Response · Authors · 2023-11-23
> > > **Thanks for the response**
> > >
> > > Thank the reviewer for the review and response again. We believe the revised ablations and applying DisenBooth to more fields may inspire some interesting future works.

---

### Official Review · Reviewer_nfwg · 2023-11-01

**Soundness:** 3 good
**Presentation:** 2 fair
**Contribution:** 3 good
**Rating:** 6
**Confidence:** 4

**Summary:**

The paper addresses a very important limitation in subject-driven T2I models - entangled representations of subject with background/irrelevant information. The key idea of the work is to learn e to learn a textual identity-preserved embedding and a visual identity-irrelevant embedding for each image containing the subject, through two novel disentangled auxiliary objectives. Experiments show superior results compared to the baselines.

**Strengths:**

The paper is very interesting and the experiment evaluation sufficiently demonstrates the utility of the proposed method. While I have few concerns in the presentation (see Weaknesses), the work could potentially have wide applicability across several applications in Gen AI space. Another attractive aspect of the paper is that the results look great with fine-tuning a small number of parameters compared to the baselines.

**Weaknesses:**

Despite multiple readings, I struggle to understand concretely the use of identity irrelevant branch in section 4.1 - which is a key proposal of the paper.  I suggest authors to expand and provide more details on this line “However, since we only need the identity-irrelevant information in this embedding, we add an adapter followed by the CLIP image encoder to filter out the identity-relevant information.” How exactly the filtering is achieved in equation (4)? Results in Figure 5 look great and show that the disentanglement is indeed working. I am happy to revise my scores after reading the rebuttal. Another question to the authors - in comparing the results with baselines such as Dreambooth, are the same number of subject images utilized? Did authors also explore fine-tuning the CLIP Text encoder? Curious to know if that would create more issues with overfitting? Overall I liked the work, results are solid - but the proposed method needs more clarity.

**Questions:**

Please see Weaknesses.

---

> ### Author Response · Authors · 2023-11-19
> **Response to Reviewer nfwg**
>
> Thank the reviewer for the constructive comments and suggestions. We have revised our paper according to all the reviewers' suggestions and uploaded the revised version, where the revised contents are colored in blue for easy reading. The reviewer can download the revised version for more qualitative results and analysis. We address your concerns point by point as follows.
>
> 1. **More details about identity-irrelevant branch**:
>
>     In the identity-irrelevant branch, we want to extract the identity-irrelevant information of the input image $x_i$. Therefore, we first resort to the pretrained CLIP image encoder $E_I$ to extract this feature, and consequently we obtain a feature $f_i^{(p)} = E_I(x_i)$. However, this feature obtained from the pretrained model may contain the information about the identity. To filter out the identity information from $f_i^{(p)}$, we design a learnable mask $M$ with the same dimension as the feature, whose element values belong to (0,1), to filter out the identity information. Therefore, we obtain a masked feature $M*f_i^{(p)}$, by the element-wise product between the mask and the pretrained feature. If one dimension of the mask is learned to 0, it means the same dimension of the pretrained feature will be discarded by multiplying 0, thus achieving the goal of filtering.
>
>     Additionally, considering that during the Stable Diffusion pretraining stage, the text encoder is jointly pretrained with the U-Net, while the image encoder is not jointly trained during Stable Diffusion pretraining, making the image feature and text feature in different space. We use the MLP with skip connection to transform $M*f_i^{(p)}$ into the same space as the text feature $f_s$,  i.e., $ f_i  =  M\*f_i^{(p)} +  MLP( M\*f_i^{(p)} ) $, which is the equation (4) in our paper as shown in the following. We call the mask and the MLP together with skip connection as the Adapter.
>
>     Thanks for your suggestion, we have revised Sec. 4.1 to make it more clear. Also, the effectiveness of the mask to filter out the identity information is validated in Appendix A.2 in our submitted version. If there is anything unclear, please feel free to let us know and we are glad to answer further questions.
>
> 2. **The number of used images**:
>
>     For fairness, we keep the number of images used by all methods the same, which is exactly the same as the dataset provided by the paper of DreamBooth. Additionally, we also provide the results of tuning with 1 image for all the methods in Appendix A.3, which further shows the superiority of our proposed method.
>
>
> 3. **Whether finetuning text encoder causes more overfitting**:
>
>     The answer is yes. Following the reviewer's advice, we try to explore the influence of tuning the text encoder for both DreamBooth and our DisenBooth, by finetuning on 20 subjects of the DreamBench. We report the average quantitative results in the following Table,
>
>     |        | DreamBooth | DreamBooth+text | DisenBooth(ours) | DisenBooth+text |
>     |--------|------------|-----------------|------------|-----------------|
>     | DINO   |      0.703 |           0.728 |      0.694 |           0.719 |
>     | CLIP-T |      0.322 |           0.286 |      0.333 |           0.301 |
>
>     where "+text" means finetuning the text encoder. We can see that both finetuning text encoder for DreamBooth and DisenBooth will increase the DINO score, meaning the generated images will be more similar to the given input images. However, the CLIP-T score (textual fidelity) will be decreased, indicating more serious overfitting problem. Despite the overfitting, we find that compared to DreamBooth, our proposed DisenBooth still shows clear superiority in reducing the impact of overfitting, surpassing DreamBooth in CLIP-T with a clear margin.
>
>     We believe this exploration is important, and we have added the results to Appendix A.4 in our revised version (Table 6), and also together with the corresponding qualitative results (Figure 12).
>
> Thanks for your suggestions, we believe the review and the revised content make the paper more clear and further improve the quality of the paper.

---

> ### Author Response · Authors · 2023-11-22
> **Gentle reminder**
>
> Dear reviewer, thank you again for the review and we hope that our response and the uploaded revised paper have addressed your concerns. As the author-reviewer is about to end, we would greatly appreciate your feedback and please feel free to let us know if you have any other questions.

---

### Comment · Area_Chair_64bH · 2023-11-20

Dear reviewers,

As the Author-Reviewer discussion period is going to end soon, please take a moment to review the response from the authors and discuss any further questions or concerns you may have.

Even if you have no concerns, it would be helpful if you could acknowledge that you have read the response and provide feedback on it.

Thanks,
AC

---

### Meta-Review · Area_Chair_64bH · 2023-12-05

**Metareview:**

The paper proposes to disentangle identity-relevant and identity-irrelevant information from the reference set to achieve better identity-preserving performance in subject-driven text-to-image generation. The main idea is to guide the fine-tuning of the diffusion model denoising process with disentangled textual identity-preserved embedding and visual identity-irrelevant embedding. The reviewers think that the proposed method has a good motivation, and is simple yet effective. The AC agrees with the reviewers and thinks that this paper indeed addresses an important problem. Therefore, an acceptance is recommended.

**Justification For Why Not Higher Score:**

While the proposed method demonstrate remarkable performance, the discussion is rather confined to a single model structure. Therefore, the generalizability is not fully verified, prohibiting it from obtaining a higher score.

**Justification For Why Not Lower Score:**

The reviewers reached a consensus of accepting this paper, and the AC agrees with them after going through the paper, reviewers, and rebuttal. Therefore, the AC decides to accept this paper.

---

### Decision · Program_Chairs · 2024-01-16

Accept (poster)